# Infant and young child feeding practice among mothers of children age 6 to 23 months in Debrelibanos district, North Showa zone, Oromia region, Ethiopia

**Mathewos Mekonnen**[ID]*[◎], **Tadele Kinati**[◎], **Kumera Bekele**[◎], **Bikila Tesfa**[◎], **Dejene Hailu**[◎], **Kemal Jemal**[ID][◎]

Department of Nursing, College of Health Sciences, Salale University, Fiche, Oromia Region, Ethiopia

◎ These authors contributed equally to this work.
* matemek2010@gmail.com

**Editor:** Julia Dratva, Zurich University of applied sciences, School of Health Professions, Leitung Forschunsstelle Gesundheitswissenschaften/Head of health sciences research, SWITZERLAND

## Abstract

### Background

Inappropriate infant and young child feeding (IYCF) practice is the leading cause of malnutrition in children. Data is needed to identify children at risk of poor feeding practice and to target interventions to improve IYCF practices. Therefore, this study aimed to assess IYCF practice and associated factors among mothers of children age 6 to 23 months in Debrelibanos district, north Showa zone, Oromia region, Ethiopia.

### Method

A community-based cross-sectional study design was conducted among 380 mothers of children age 6 to 23 months from March 1 to April 5, 2019. A simple random sampling technique was used to select the respondents. Data was collected using a structured interviewer-administered questionnaire that had been pretested. The data was entered into Epi-Data 3.1 and then transferred to SPSS 21 for analysis. Descriptive statistical analysis was done, and an association between an outcome variable and independent variables was examined in logistic regression models.

### Result

Overall, 65.8% of mothers practiced appropriate IYCF practice. The study revealed that 70.5% of children started breastfeeding within one hour of birth, and 61.6% were breastfed exclusively for six months. Among studied mothers, 79.5% continued to breastfeed their children until 2 years, and 69.2% of the participants started complementary feeding timely at six months. Minimum dietary diversity was observed in 19.2% of children, while minimum meal frequency was found in 79.2%. The majority of mothers (77.6%) fed their babies with bottles. Mother's educational status of primary school [AOR = 4.50, 95% CI: (1.38,14.61)], husband's occupation being merchant [AOR = 6.45, 95% CI: (1.51, 27.59)]; antenatal care follows up [AOR = 3.15, % CI: (1.22, 8.12)], radio/television ownership [AOR = 7.41, 95%

**Data Availability Statement:** All relevant data are within the paper and its Supporting Information files.

**Funding:** This study was funded by Salale University. The funder had no role in study design, data collection, and analysis, decision to publish, or preparation of the manuscript.

**Competing interests:** The authors declare that no competing interests exist.

CI: (2.86, 19.20)], child's sex being female [AOR = 4.78, 95% CI: (2.26, 10.064) and sufficient knowledge on child feeding [AOR = 2.82, 95% CI: (1.27, 26.26)] were independent predictors for appropriate IYCF practice.

## Conclusion

The prevalence of appropriate infant and young child feeding practice indicators was found to be rather high among the mothers in this study. The use of a bottle to feed babies, in particular is very common among the mothers who were studied. To address child malnutrition, it is critical to educate families about proper IYCF practices. This study suggests that mothers be properly educated about IYCF recommendations at health care facilities during their visits, as well as the promotion of appropriate IYCF through various media.

## Introduction

Malnutrition accounts for around two-thirds of all deaths among children under the age of five worldwide. Each year, more than 10 million children die as a result of malnutrition; 98% of these deaths occur in underdeveloped countries [1]. Although the risk of death is higher in stunted children, it is exacerbated when wasting and stunting occur at the same time [2]. Children's health, physical growth, cognitive development, and school performance are all compromised by chronic malnutrition [3]. Stunting in childhood has also been linked to a higher risk of death for mothers in the future, as well as poor pregnancy outcomes such as preterm and low birth weight child [4, 5]. Furthermore, it has a detrimental impact on adults' economic productivity as well as the country's overall gross domestic product [6]. In Ethiopia, according to 2016 Ethiopian Demographic Health Survey (EDHS) 38 percent and ten percent of children under the age of five are stunted and wasted, respectively [7]. Undernutrition is rampant in Ethiopia, owing primarily to inappropriate feeding practice [8].

World Health Organization (WHO) and the United Nations Children's Education Fund (UNICEF) have demonstrated that basic nutrition programs can avert a quarter of child mortalities and one-third of chronic malnutrition. As a result, UNICEF and WHO have established a global strategy for optimal IYCF to combat child malnutrition and mortality [9]. Early initiation of breastfeeding; exclusive breastfeeding during the first 6 months of life; timely introduction of complementary feeding at six months; continued breastfeeding after complementary foods is introduced; adequate dietary diversity in complementary foods; adequate frequency of foods and nonuse of bottle for feeding baby are important aspects of IYCF in the first 2 years of life [10–12].

According to the 2016 EDHS, 58 percent of children were exclusively breastfed during the first six months of their lives. At two years, 76 percent of mothers continued to breastfeed their infants, while 60 percent of mothers began supplemental feeding on time. Furthermore, 14 percent and 45 percent of mothers, respectively, had a minimum dietary and meal frequency [7].

Efforts to enhance the nutritional status of children in Ethiopia have been made at various times, including the adoption of IYCF recommendations [13]. However, these attempts have been ineffective in bringing about significant and long-term changes in IYCF practices. In a study conducted in Shashamane, Ethiopia, for instance, the prevalence of inappropriate IYCF practice was found to be 67.9% [14].

To build programs that could direct and impact public health initiatives, combined data on key elements of IYCF practice is required. Previous studies on child feeding, on the other hand, mainly focused on child feeding components separately. There are very few studies on the IYCF index's combined variables based on WHO-identified key indicators in Ethiopia. As a result, this study aimed to assess IYCF practice and associated factors among mothers of children age 6 to 23 months in Ethiopia's Debrelibanos district, North Showa zone, Oromia region.

## Methods

### Study design, period, and area

From March 1 to April 5, 2019, a community-based cross-sectional study was undertaken. The study was conducted in Ethiopia's Oromia region, in the Debrelibanos district of the North Showa zone. The district is 90 kilometers from Ethiopia's capital, Addis Ababa. One urban and ten rural kebeles make up the district. The population of the Debrelibanos district was estimated to be 64,305, according to unpublished figures acquired from the district. The ratio of males to females was 1:1. The reproductive age group (15–45 years) accounts for 22.23 percent of the female population, with 3,672 women having children under the age of two years.

### Study population

The study population for this study was all mothers with children age 6 to 23 months who had lived in the district for more than six months. Mothers who were critically ill and unable to communicate were excluded from the study.

### Sample size and sampling technique

The sample size for this study was calculated using a formula for a single population proportion, which took into account the power of 80%, the confidence interval level of 95%, the margin of error of 5%, and the prevalence of appropriate IYCF practice 32.1% [14]. The sample size was 385 after adjusting for the non-response rate. Due to the small number of *kebeles* in the district, all of them were considered. The sample size was then proportionally allocated to the size of each population size of *kebele*. Finally, the participants were selected using simple random sampling.

### Data collection tools and procedure

The instrument was developed using previous literature, national guidelines, and the WHO's IYCF practice guideline, and it was pretested before actual data collection. Mother and child demographic characteristics, household characteristics, maternal and child health care service use, paternal involvement in child feeding, knowledge of IYCF, an attitude of mothers toward IYCF, and IYCF practices were all included in the questionnaire. To assess IYCF practice, seven WHO indicators were used. For each indicator, the appropriate feeding practice was assessed based on compliance with WHO recommended practices. A 24-hour recall approach was used to assess minimum dietary diversity, minimum meal frequency, continued breastfeeding, and bottle feeding. However, a child's lifetime historical recall approach was used to elicit early initiation of breastfeeding, exclusive breastfeeding, and timely initiation of complementary feeding. Under the supervision of two BSc nurses, face-to-face interviews by eight female diploma nurses were conducted. The data collectors and supervisors received training on the study's purpose and data collection methodology.

## Operational definitions

**Early initiation of breastfeeding**: initiation of breast milk within one hour after delivery

**Exclusive breastfeeding**: receiving only breast milk, except drops or syrup consisting of vitamins, minerals, or medicines up to six months.

**Timely initiation of complementary feeding**: introducing solid or semi-solid food at 6–8 months

**Minimum dietary diversity**: the proportion of children 6–23 months of age who consume four or more out of seven standard groups. Food groups include grains, roots, and tubers; legumes and nuts; dairy products; flesh foods, eggs; vitamin A-rich fruits and vegetables; other fruits and vegetables [15].

**Minimum meal frequency**: children age 6–23 months who receive solid, semi-solid, or soft foods the minimum number (6–8 months, 2 times; 9–23 months, 3 times for breastfed children and 6–23 months, 4 times for non-breastfed children) [15].

**Continued breastfeeding**: Proportion of children 6–23 months of age who received breast milk

**Bottle feeding:** Proportion of children 6–23 months of age who are fed with a bottle.

**Appropriate IYCF practice:** was measured from seven WHO indicators (i.e., early initiation of breastfeeding, exclusive breastfeeding, no bottle feeding, minimum meal frequency, minimum dietary diversity, timely introduction of complementary foods, and continued breastfeeding in children 6–23 months). If a child fulfilled at least mean or four of the seven criteria, it was classified as appropriate IYCF practice [11, 14].

**Knowledge about IYCF**: was measured from seven knowledge questions about child feeding. Based on the summation score, a score above 50% was considered as having sufficient knowledge about IYCF

## Data processing and analysis

Completeness and consistency of responses were checked on filled questionnaires. The data was manually pre-coded, then entered into Epi-Data version 3.1 statistical software, which was then transferred to SPSS 21 for analysis. Data was edited by checking for missing values and outliers after running frequencies.

During analysis the age-appropriate IYCF indicator questions were recoded into the same variables and coded as (yes = 1 and no = 0) to make the variable dichotomous, then the coded response was computed to get the mean response of seven IYCF indicator questions, after that the mean response of the seven IYCF questions was computed to get the overall mean response of seven IYCF practice, then those who responded above the mean were coded as (1 appropriate IYCF practice and 0 for inappropriate IYCF practice).

To measure knowledge on IYCF; seven knowledge questions recoded as (yes = 1 and no = 0) were used to construct a composite score. Based on the summation score, a score above 50% was considered as having sufficient knowledge about IYCF

The results of a descriptive statistical analysis were given in percentages, means, standard deviations, using tables. Binary logistic regression was used to perform bivariate analysis between dependent and independent variables. The P-value of $\leq 0.25$ was taken as a cut-off point to select eligible variables for the multiple logistic regression analysis to control for potential confounders. Association was expressed as crude odds ratios (COR) and adjusted odds ratio (AOR), and their corresponding 95% confidence intervals (CIs) were obtained from logistic regression models. Odds ratios (OR) were reported together with their 95% CI, and statistical significance was deemed at P-value $< 0.05$.

### Ethical considerations

Salale University Ethical Review Committee provided ethical approval for this study. The district issued a permission letter. Mothers gave their written/ fingerprint consent. The participants were told that they had the freedom to participate or not. Participants were encouraged to be as honest as possible because their information is valuable to the district, region, and country as a whole. By omitting their names from the questionnaire, the participants' confidentiality was preserved

## Result

### Maternal socio-demographic characters

A total of 380 mothers with children age between 6 to 23 months participated in the study with a response rate of 98.7%. The mean age of the mothers was 28.36 (±SD = 5.78) years. The majority of the participants 308 (81.1%) were housewives /farmers, with an average monthly income of nearly 1,000 Ethiopian Birr (ETB). The Orthodox made up 339 (89.2%), while the Protestants made up 36 (9.5%). About a third of the respondents (34.7%) couldn't read or write (Table 1).

### Maternal, child, and household characteristics

Three-quarters of the mothers in the study, 287 (75.5%), were multipara, and approximately half of the families, 203 (53.4%), had one or two children. Males account for half (50.8%) of the children who participated in the study. The majority of the children in the study, 267 (64.5%), were between the ages of 12 and 23 months. The mean of children was 13.43 months (SD = 5.216) at the time of the study. A significant number of husbands, 149 (39.2%), were not involved in child feeding. More than three-quarters of the mothers 263, (69.2%) had sufficient knowledge on IYCF practice. Two hundred thirty-four (61.6%) of mothers had a positive attitude towards IYCF practice (Table 2).

### Infant and young child feeding practices

Nearly two-thirds of mothers (65.8%) practiced appropriate IYCF. In terms of IYCF components, the majority of mothers, 268 (70.5%), reported starting breastfeeding within one hour of birth, and 234 (61.6%) gave their child solely breast milk for the first six months. A considerable number of mothers 295 (77.6%) fed their babies with a bottle. At six months, over two-thirds of the mothers 263(69.2%) started complementary feeding timely. Only 73 (19.2%) of the respondents met the minimum dietary diversity requirements. The majority of mothers, 302 (79.5%), continued breastfeeding, and the minimum meal frequency was 301 (79.2%) for complementary food (Table 3).

### Factor associated with infant and young infant feeding practices

In binary logistic regression model, age of mother, educational status of the mother, occupation of mother, husband's educational status, husband's occupation, family radio/television ownership, mother's parity, ANC follow up, place of delivery, sex of the child, paternal involvement in child feeding, number of children in the household, knowledge on IYCF and attitude towards IYCF practices were significantly associated with IYCF practices (Table 4).

In multiple logistic regression, mothers with primary school educational status were more likely to practice appropriate IYCF practices than odds of illiterate mothers [AOR = 4.50, 95% CI: (1.38, 14.61)]. A mother whose husband was a merchant was more likely to practice appropriate feeding than a mother whose husband was a farmer [AOR = 6.47, 95% CI: (1.51, 27.59)].

**Table 1. Socio-demographic characteristics of the study participants at Debrelibanos district, north Showa, Oromia, and Ethiopia.**

| Variables | | Frequency | Percent |
|---|---|---|---|
| Age of mother | 15–24 | 88 | 23.2 |
| | 25–34 | 216 | 56.8 |
| | 35–45 | 76 | 20.0 |
| Marital Status | Married | 346 | 91.0 |
| | Divorced | 22 | 5.8 |
| | Other [a] | 12 | 3.2 |
| Religion | Orthodox | 339 | 89.2 |
| | Protestant | 36 | 9.5 |
| | Others [b] | 5 | 1.3 |
| Mother's educational status | Unable to read & write | 132 | 34.7 |
| | Able to read &write | 25 | 6.6 |
| | Primary school | 87 | 22.9 |
| | Secondary school | 97 | 25.5 |
| | College and above | 39 | 10.3 |
| Mother's occupation | Housewife/farmer | 308 | 81.1 |
| | Government employee | 35 | 9.2 |
| | Merchant | 30 | 7.9 |
| | Others [c] | 7 | 1.8 |
| Husband's education status | Unable to read & write | 122 | 32.1 |
| | Able to read &write | 14 | 3.7 |
| | Primary school | 80 | 21.0 |
| | Secondary school | 93 | 24.5 |
| | College and above | 71 | 18.7 |
| Husband's occupation | Farmer | 253 | 66.6 |
| | Merchant | 54 | 14.2 |
| | Government employee | 51 | 13.4 |
| | Others [d] | 22 | 5.8 |
| Family income in ETB | < = 1000 | 183 | 48.2 |
| | 1001–2000 | 74 | 19.5 |
| | 2001–3000 | 40 | 10.5 |
| | > = 3001 | 83 | 21.8 |
| Radio/TV ownership | No | 123 | 32.4 |
| | Yes | 257 | 67.6 |

[a] single and widowed

[b] Muslim and wakefata

[c] private employee and daily laborer

[d] private employee and daily laborer; ETB, Ethiopian Birr.

Mothers from the family who had radio/television at home were seven-time more likely to practice proper feeding practices [AOR = 7.41, 95% CI: (2.86, 19.20)]. Mothers who made ANC follow up during pregnancy were three times more likely to practice appropriate IYCF practices [AOR = 3.15, 95% CI: (1.22, 8.12)]. Mothers who had sufficient knowledge of IYCF were more likely to practice appropriate feeding practices [AOR = 2.82, 95% CI: (1.27, 26.26)]. Finally, having a female child was found to be a predictor for appropriate IYCF practices [AOR = 4.78, 95% CI: (2.26, 10.06)] (Table 4)

**Table 2. Maternal, child, and household characteristics of the study participants, Debrelibanos district, north Showa, Oromia, Ethiopia.**

| Variables | | Frequency | Percent |
|---|---|---|---|
| Parity | Primpara | 93 | 24.5 |
| | Multipara | 287 | 75.5 |
| ANC follow up | No | 69 | 18.2 |
| | Yes | 311 | 81.8 |
| Place of delivery | Home | 111 | 29.2 |
| | Health facility | 269 | 70.8 |
| Current pregnancy status | No | 338 | 88.9 |
| | Yes | 42 | 11.1 |
| Sex of child | Male | 193 | 50.8 |
| | Female | 187 | 49.2 |
| Age of child | 6–8 | 81 | 21.3 |
| | 9–12 | 57 | 15.0 |
| | 12–23 | 242 | 63.7 |
| Number of children in the family | 1–2 | 203 | 53.4 |
| | 3–4 | 117 | 30.8 |
| | > = 5 | 60 | 15.8 |
| Paternal involvement in child feeding | No | 149 | 39.2 |
| | Yes | 231 | 60.8 |
| Mother's knowledge on IYCF | Insufficient | 117 | 30.8 |
| | Sufficient | 263 | 69.2 |
| Mother's attitude towards IYCF | Negative | 146 | 38.4 |
| | Positive | 234 | 61.6 |

ANC, Antenatal care; IYCF, infant and young child feeding.

**Table 3. Infant and young child feeding practice components among mothers with children age between 6 to 23 months at Debrelibanos district, north Showa, Oromia, Ethiopia.**

| IYCF Component | | Frequency | Percent |
|---|---|---|---|
| Early initiation of breastfeeding | Yes | 268 | 70.5 |
| | No | 112 | 29.5 |
| Exclusive breastfeeding | Yes | 234 | 61.6 |
| | No | 146 | 38.5 |
| No bottle-feeding | Yes | 85 | 22.4 |
| | No | 295 | 77.6 |
| Timely initiation of complementary food | Yes | 263 | 69.2 |
| | No | 117 | 30.8 |
| Minimum dietary diversity | Yes | 73 | 19.2 |
| | No | 307 | 80.8 |
| Minimum meal frequency | Yes | 301 | 79.2 |
| | No | 79 | 20.8 |
| Continued breastfeeding | Yes | 302 | 79.5 |
| | No | 71 | 20.5 |
| IYCF practice | Appropriate | 250 | 65.8 |
| | Inappropriate | 130 | 34.2 |

IYCF, infant and young child feeding.

**Table 4. Binary and multiple variables logistic regression analysis of factors associated with IYCF practices among mothers with children age between 6 to 23 months at Debrelibanos district, north Showa, Oromia, Ethiopia.**

| Variables | Category | Appropriate IYCF Practice | | COR (95%CI) | AOR (95%CI) |
|---|---|---|---|---|---|
| | | No (N) | Yes (N) | | |
| Age of mother | 15–24 | 23 | 65 | 1 | |
| | 25–34 | 70 | 146 | 0.74 (0.42, 1.29) | |
| | 35–45 | 37 | 39 | 0.37 (0.19, 0.72) | |
| Mother's education | Unable to read & write | 82 | 50 | 1 | 1 |
| | Able to read &write | 3 | 22 | 12.03 (3.42, 42.25) | 1.31 (.232, 7.36) |
| | Primary School | 10 | 77 | 12.63 (5.98, 26.65) | **4.50 (1.38, 14.61)** * |
| | Secondary School | 25 | 72 | 4.72 (2.66, 8.40) | 1.12 (0.31, 4.10) |
| | College and above | 10 | 29 | 4.76 (2.14, 10.59) | 0.58 (0.12, 2.81) |
| Mother's occupation | Housewife/Farmer | 118 | 190 | 1 | |
| | Government employee | 7 | 28 | 2.48 (1.05, 5.87) | |
| | Merchant | 3 | 27 | 5.59 (1.66, 18.83) | |
| | Others [a] | 2 | 5 | 1.55 (0.30, 8.13) | |
| Husband's education | Unable to read & write | 82 | 40 | 1 | |
| | Able to read &write | 2 | 12 | 12.30 (2.62, 57.60) | |
| | Primary school | 11 | 69 | 12.86 (6.14, 26.96) | |
| | Secondary school | 16 | 77 | 9.87 (5.11, 19.05) | |
| | College and above | 19 | 52 | 5.61 (2.94, 10.72) | |
| Husband's occupation | Farmer | 110 | 143 | 1 | 1 |
| | Merchant | 6 | 48 | 6.15 (2.54, 14.90) | **6.45 (1.51, 27.59)** * |
| | Government employee | 12 | 39 | 2.50 (1.25, 5.00) | 0.99 (0.35, 2.81) |
| | Others [a] | 2 | 20 | 7.69 (1.76, 33.61) | |
| Radio/TV ownership | No | 89 | 34 | 1 | 1 |
| | Yes | 41 | 216 | 13.79 (8.22, 23.13) | **7.41 (2.86, 19.20)** * |
| Parity | Primary | 17 | 76 | 1 | |
| | Multipara | 113 | 174 | 0.34 (0.19, 0.61) | |
| ANC follow up | No | 44 | 25 | 1 | 1 |
| | Yes | 86 | 225 | 4.60 (2.66, 7.98) | **3.15 (1.22, 8.12)** * |
| Place of delivery | Home | 61 | 50 | 1 | |
| | Health facility | 69 | 200 | 3.54 (2.23, 5.62) | |
| Current pregnancy status | No | 119 | 219 | 1 | |
| | Yes | 11 | 31 | 1.53 (0.74, 3.16) | |
| Sex of child | Male | 83 | 110 | 1 | 1 |
| | Female | 47 | 140 | 2.25 (1.45, 3.48) | **4.78 (2.26,10.06)** * |
| Age of child | 6–8 | 26 | 55 | 1 | |
| | 9–12 | 26 | 31 | 0.56 (0.28, 1.13) | |
| | 12–23 | 78 | 164 | 0.99 (0.58, 1.70) | |
| Number of children in the family | 1–2 | 53 | 150 | 1 | |
| | 3–4 | 54 | 63 | 0.41 (0.26, 0.67) | |
| | > = 5 | 23 | 37 | 0.57 (0.31, 1.04) | |
| Paternal involvement | No | 76 | 73 | 1 | |
| | Yes | 54 | 177 | 3.41 (2.19, 5.31) | |
| Knowledge on IYCF | Insufficient | 72 | 45 | 1 | 1 |
| | Sufficient | 58 | 205 | 5.66 (3.52, 9.08) | **2.82 (1.27, 6.26)** * |

(*Continued*)

**Table 4.**  (Continued)

| Variables | Category | Appropriate IYCF Practice | | COR (95%CI) | AOR (95%CI) |
|---|---|---|---|---|---|
| | | No (N) | Yes (N) | | |
| Attitude towards IYCF | Negative | 69 | 77 | 1 | |
| | Positive | 61 | 173 | 2.54 (1.64,3.93) | |

ANC, Antenatal care; IYCF, infant and young child feeding; TV, Television; COR; Crude odds ratio; AOR, Adjusted odds ratio

*Shows significant association for multivariate logistic regressions at 95% CI

## Discussion

In order to reduce malnutrition in a developing country like Ethiopia, safe, adequate and acceptable infant and child feeding practice is vital. For this reason, WHO and UNICEF have recommended core infant feeding practices. This study assessed IYCF practice and associated factors among mothers of children age 6 to 23 months using seven WHO indicators. In the current study, a significant number of mothers practiced appropriate IYCF practice. Furthermore, although, the practice of bottle feeding is extremely high, the practice of recommended feeding practices is relatively better as compared to the national prevalence and other previous studies' report in majority of IYCF practice components. Mother's educational status of being primary school, husband's occupation being merchant, ANC follow up, possession of radio/television, child's sex being female and sufficient knowledge on child feeding were independent positive predictors of appropriate IYCF practice.

The current study's prevalence of appropriate IYCF practice (65.8%) is higher than a study conducted in Ethiopia's Shashamane Woreda, where 32.1% of caregivers practiced appropriate IYCF practice [14]. The inconsistency could be due to the length of time between studies, and in Ethiopia, the number of mothers seeking maternal health care has increased considerably as a result of the country's ongoing promotion of free maternal services, providing a favorable chance for health professionals to promote IYCF practice. Even though the WHO guideline on IYCF practice [16] does not specify minimum standards to be met or the percentage that should be considered for public health importance, it is understandable that all children (6–23 months) should follow all of the recommended feeding practices.

Early initiation of breastfeeding is important for both the mother and the child. As a result, it is suggested that children be put to the breast as soon as possible after birth, preferably within one hour of birth. In this study, a little more than two-thirds of mothers (70.5%) started breastfeeding within one hour of delivery. The finding is nearly identical to the national prevalence (73%) according to the 2016 EDHS [7]. The result is, however, slightly lower than that of a study conducted in Mekelle town, Ethiopia (78%) [17]. This could be due to the fact that the previous study was conducted in a town where healthcare and the media are more readily available than in the current study area, where the majority of the study population comes from rural areas.

The prevalence of exclusive breastfeeding was 61.6% in this study. This finding is comparable to what was found at the national level (58%) in the EDHS 2016 and Mekelle (60.8%) [7, 17]. However, it is lower than another study conducted in Ethiopia (87.8%) [14]. The discrepancy could be attributed to differences in exclusive breastfeeding measurement, where the former study assessed exclusive breastfeeding based on 24-hour recall, and the current study assessed retrospectively asking about exclusive breastfeeding.

The introduction of nutritionally adequate and safe complementary foods to infants and young children enhances growth and development. According to the finding, 69.2% of the

children studied received complementary foods timely. This finding is in line with reports from developing countries such as Nepal (70%) [18] and Bangladesh (71%) [19]. However, the finding is lower when compared to other African countries, including Uganda (75%), Tanzania (92.3%), and Kenya (81%) [20–22] but higher than other studies conducted in Ethiopia (51–62.8%) [23–25], India (55.1%) [26] and national prevalence as per EDHS, 2016 (60%) [7]. Disparity across the studies in reporting time of initiating complementary food could be due to the use of different methods of measurement time of initiation: at the sixth-month or using reference period 6–8 months.

According to WHO's IYCF recommendation, breastfeeding should be continued for the first two years, along with complementary food. The current study found that 79.5% of the mothers in the study continued to breastfeed their children. This finding is consistent with the national prevalence in Ethiopia (76%) [7]. It is, however, higher than the 43.3% reported in an Indian study [26]. This could be due to the Ethiopian context's intensified health education package provided by health extension workers.

Dietary diversity is a proxy for adequate micronutrient density of foods. Food intake from at least four of the seven food groups is assessed in children aged 6 to 23 months as minimum dietary diversity. This study's minimum food diversity (19.2%) was greater than the national prevalence (14%) [7] and a study in the Gorche district of southern Ethiopia (10.6%) [27]. However, it was lower than the findings of studies conducted in Bangladesh (41.5%) and Sri Lanka (71.1%) [19, 28]. This could be due to a lack of availability to the required food combination.

Minimum meal frequency examines the number of times children received foods other than breast milk. The minimum number is specific to the age and breastfeeding status of the child. Accordingly, the minimum meal frequency in this study (79.2%) is almost similar to India (78%) [26] and Bangladesh (81.1%) [19]. However, it is lower than Sri Lanka (88.3%) [28] and Ethiopia (45%) according to 2016 EDHS [7].

Using a bottle with a nipple to feed an infant, a practice that is discouraged increases the child's risk of illness and decreases the child's desire to breastfeed, perhaps resulting in a decrease in milk output. The majority of mothers (77.6%) used bottles to feed their babies, according to the findings of the study. This prevalence is higher than that reported by the EDHS in 2016 (14%) (7), Western Uganda (10%) [20], and a study conducted in rural Ethiopia (39.8%) [29]. The highest finding could be attributed to the area's highest milk production, and cow's milk is more convenient to provide to children using bottles than other foods.

There was an association between ANC follow-up and appropriate IYCF practices among mothers in this study. This result is supported by research from Ethiopia [14] and Nepal [30]. The association may be due to the fact that mothers who had ANC visits may have received related health information from their health care providers. As a result, ensuring that pregnant mothers receive recommended ANC could be a useful strategy for improving IYCF practice.

In the current study, having a radio or television had a significant relationship with appropriate IYCF practice. The result is supported by secondary data analysis from the EDHS and another study from Northwest Ethiopia [31, 32]. Media ownership was also identified as a key driver of IYCF practice in an Indian study [33]. Because the media is generally regarded as a trustworthy source of health and nutrition information and such messages are more likely to be implemented. Mothers who have radio are also more likely to be exposed to IYCF education provided through mass media.

Appropriate infant and young child feeding practice was found to be significantly associated with having a female child in the current study, with females four times more likely to be appropriately fed than their male counterparts. This result is consistent with a study conducted in the Ethiopian town of Asella and India, which found that being female, was positively related to receiving the proper IYCF practice [34]. Some authors [35] speculated that women

may believe that boy infants have greater nutritional needs and hence require complementary feeding at an earlier age, resulting in an untimely introduction of complementary food and therefore affecting exclusive breastfeeding practice. Male children are weaned earlier than female children, according to another Algerian study [36]. As a result, mothers must be educated that breast milk alone is adequate to provide essential nutrition for infants' optimal growth and development during the first six months of life, and that breastfeeding should be continued until the age of two years, irrespective of sex.

The mother's educational status was associated with appropriate IYCF practice. This result is consistent with findings from prior studies in Nepal, Pakistan, and Ethiopia [37–39]. This could be because educated women grasp nutrition education better than less educated or uneducated mothers. Furthermore, educated mothers are more likely than their counterparts to read books, pamphlets, and periodicals, as well as be exposed to nutrition education concerning IYCF via mass media. Hence mothers with a lower level of education should be given extra support to help maintain to practice appropriate IYCF practice.

Being a merchant is positively associated with appropriate IYCF practice when it comes to the husband's occupation. This could be related to the fact that merchants in the study area have a relatively high income, which could lead to increased purchasing power and easier access to a variety of foods as they begin complementary food, as opposed to other occupations.

Appropriate IYCF practice was associated with having sufficient knowledge on IYCF practice. Another study in rural Ethiopia [40] came up with similar results. The likely reason could be linked to the importance of knowledge in empowering mothers to resist external pressures and interferences from traditional views and misconceptions about IYCF. The study finding could support reinforcing nutrition education through tailoring with behavioral change and communication components to address the community misunderstanding towards IYCF practice. As a result, efforts to improve mothers' IYCF knowledge should be escalated.

There are some limitations to this study: The study's cross-sectional design, for example, makes it difficult to assess the cause-and-effect link between possible causes and the outcome. Because of recall and social desirability biases, the 24-hour recall approach may result in an overestimation of the proportion of IYCF practices. Discrepancies in the measurement of feeding practices, as well as the real meaning of each practice, also make building and interpreting composite feeding practices challenging. Another limitation of the study is its generalizability; the population was sampled from only one Woreda, thus it may not be representative of the region or country.

## Conclusion

In conclusion, the prevalence of appropriate infant and young child feeding practices indicators was found to be rather high among the mothers in this study. Bottle feeding in particular is, very common among the women who were studied. Mother's educational status, husband's merchant occupation, family radio/television ownership, ANC follow-up, child's sex, and sufficient knowledge of IYCF were all independent predictors of appropriate IYCF practice. In order to combat child malnutrition, it is critical to educate mothers about appropriate IYCF practices. This study suggests that health workers educate mothers about IYCF guidelines during antenatal care, particularly for mothers with lower educational levels; it also recommends that appropriate IYCF be promoted through mass media.

## Supporting information

**S1 Data. Data used in the study.**
(SAV)

**S1 File. Questionnaire.**
(DOCX)

## Acknowledgments

The authors thank Salale University for its unreserved support, follow-up, and sponsorship for this study. We are also grateful to Debrelibanos district and respective *Kebele* leaders for their assistance during data collection.

## Author Contributions

**Conceptualization:** Mathewos Mekonnen, Tadele Kinati, Kumera Bekele, Bikila Tesfa, Dejene Hailu, Kemal Jemal.

**Data curation:** Mathewos Mekonnen, Kumera Bekele, Dejene Hailu.

**Formal analysis:** Mathewos Mekonnen, Tadele Kinati, Bikila Tesfa, Kemal Jemal.

**Funding acquisition:** Mathewos Mekonnen.

**Investigation:** Mathewos Mekonnen, Tadele Kinati, Kumera Bekele, Bikila Tesfa, Dejene Hailu.

**Methodology:** Mathewos Mekonnen, Tadele Kinati, Kumera Bekele, Bikila Tesfa, Dejene Hailu, Kemal Jemal.

**Project administration:** Mathewos Mekonnen.

**Resources:** Mathewos Mekonnen.

**Software:** Mathewos Mekonnen.

**Supervision:** Mathewos Mekonnen, Kumera Bekele, Dejene Hailu.

**Validation:** Mathewos Mekonnen.

**Visualization:** Bikila Tesfa, Dejene Hailu, Kemal Jemal.

**Writing – original draft:** Mathewos Mekonnen, Kemal Jemal.

**Writing – review & editing:** Mathewos Mekonnen, Tadele Kinati, Kumera Bekele, Bikila Tesfa, Dejene Hailu, Kemal Jemal.

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
