## [Decision Letter · Decision Letter 0]

19 Apr 2021

PONE-D-20-34804

Infant and Young Child Feeding Practice among Mothers with Children age between 6 to 23 Months at Debrelibanos District, North Showa, Oromia Ethiopia

PLOS ONE

Dear Dr. Gemmechu,

Thank you for submitting your manuscript to PLOS ONE. After careful consideration, by one external reviewer and the academic editor, we feel that it has merit but does not fully meet PLOS ONE’s publication criteria as it currently stands. Therefore, we invite you to submit a revised version of the manuscript that addresses the points raised during the review process.

We look forward to receiving your revised manuscript.

Kind regards,

Julia Dratva

Academic Editor

PLOS ONE

Journal Requirements:

Additional Editor Comments:

Dear Authors

The manuscript touches on an important topic Infant and Young Child Feeding (IYCF) in Ethopia. It investigates the prevalence of appropriate parctice and factors associated with appropriate practice to improve communication and practice in future.

While the paper presents interesting results and compares them with former research on the topic, the results are insufficiently presented and the comaprisons lack information on the former studies. Overall, the statistical methods can be presented better, to allow readers to interprete results better. The paper needs an English editing and a major revision.

Specific points:

Some sentences are not understandable and can be misinterpreted, e.g. "Mothers who were critically ill and not volunteer to participate in the study were excluded." Are these 2 exclusion criteria or one, mothers critically ill not volonteering....

In general non-participation is not an exclusion criterium.

Authors should explain if the district chosen to study IYCF is representative of the country as a whole and if the results are generalizable.

24 hour recall is mentioned among the operational definitions. It would be worthwhile to present the method in the data colelciton part and address how these data were analyzed.

Appropriate IYCF practices - is the threshold a standard threshold - no reference was cited. Please elaborate.

Similarly, please define "sufficient knowledge" when was it considered sufficient.

Statistical methods: please add the confounding factors used in the multivariate regression analyses, or was this rather a explorative approach with all co-variates adjusting each other?

Discussion: Authors prsent other studies on IYCF - however only vaguely refer to explanations for differences - such as might be due to socio-economic differences. This is not satisfying for the readers. Elaborate.

the two least well practiced recommendations (diversity in foods, bottle-feeding) merit more discussion, also due to the question of availability/access to diverse foods. In this case, education on IYCF success seem limeted.

The discussion should also include a section on generalizability.

Minor points:

Tbl. 1: familiy income - <1000 is twice

Gender bias, to me this does not seem to be a bias, but a gender difference in feeding practices. There is some literature on the topic and hypothesis on the reasons for this difference.

4. Please amend the manuscript submission data (via Edit Submission) to include author Tadele Kinati, Kumera Bekele, Bikila Tesfa, Dejene Hailu, and Kemal Jemal.

Reviewers' comments:

Reviewer's Responses to Questions

**Comments to the Author**

1. Is the manuscript technically sound, and do the data support the conclusions?

Reviewer #1: Partly

2. Has the statistical analysis been performed appropriately and rigorously? 

Reviewer #1: No

3. Have the authors made all data underlying the findings in their manuscript fully available?

Reviewer #1: No

4. Is the manuscript presented in an intelligible fashion and written in standard English?

Reviewer #1: No

5. Review Comments to the Author

Reviewer #1: I found that title of the manuscript was very interesting. However, the whole body of the manuscript was written in poor English. Even the language in the abstract seems a zero-draft document. The discussion is very shallow which leads to shallow conclusion. The binary and multivariate analysis should have been separately presented. Nothing was stated on how the candidate variables for the multivariate analysis were selected. What's the justification behind including all the 11 kebeles of the district? In general, this manuscript needs critical revision.

6. PLOS authors have the option to publish the peer review history of their article (what does this mean?). If published, this will include your full peer review and any attached files.

Reviewer #1: No

---

## [Author Response · Author response to Decision Letter 0]

3 Jun 2021

Journal Requirements:

 Reply: Thank you for the guidance, the PLOS ONE’s style requirement have been incorporated accordingly in all sections.

 Reply: Based on your advices the whole manuscript has been copy edited thoroughly for language usage, spelling and grammar by one of our colleagues not listed as author, who is expert in English. 

 Reply: The corresponding author has already ORCID iD (0000-0002-3794-7818)

4. Please amend the manuscript submission data (via Edit Submission) to include author Tadele Kinati, Kumera Bekele, Bikila Tesfa, Dejene Hailu, and Kemal Jemal.

 Reply: The manuscript submission data has been amended accordingly 

 Reply: Modified accordingly 

Additional Editor Comments:

Dear Authors

The manuscript touches on an important topic Infant and Young Child Feeding (IYCF) in Ethopia. It investigates the prevalence of appropriate parctice and factors associated with appropriate practice to improve communication and practice in future.

While the paper presents interesting results and compares them with former research on the topic, the results are insufficiently presented and the comaprisons lack information on the former studies. Overall, the statistical methods can be presented better, to allow readers to interprete results better. The paper needs an English editing and a major revision.

Reply: The result has been sufficiently presented and comparison on previous studies have been made thoroughly. The methodology has been presented in the better way so that it can be understood easily by reader. The language is copy edited 

Specific points:

Some sentences are not understandable and can be misinterpreted, e.g. "Mothers who were critically ill and not volunteer to participate in the study were excluded." Are these 2 exclusion criteria or one, mothers critically ill not volonteering....

In general non-participation is not an exclusion criterium.

Reply: Modification has been made accordingly 

Authors should explain if the district chosen to study IYCF is representative of the country as a whole and if the results are generalizable.

Reply: The issue of generatability has been discussed generously as limitation 

24 hour recall is mentioned among the operational definitions. It would be worthwhile to present the method in the data colelciton part and address how these data were analyzed.

Reply: Modification has been made by authors accordingly 

Appropriate IYCF practices - is the threshold a standard threshold - no reference was cited. Please elaborate.

Similarly, please define "sufficient knowledge" when was it considered sufficient.

Reply: Modifications have been made by authors accordingly 

Statistical methods: please add the confounding factors used in the multivariate regression analyses, or was this rather a explorative approach with all co-variates adjusting each other?

Reply: The P value ≤0.25 was taken as a cut-off point to select eligible variables for the multiple logistic regression analysis to control for potential confounders.

Discussion: Authors present other studies on IYCF - however only vaguely refer to explanations for differences - such as might be due to socio-economic differences. This is not satisfying for the readers. Elaborate.

the two least well practiced recommendations (diversity in foods, bottle-feeding) merit more discussion, also due to the question of availability/access to diverse foods. In this case, education on IYCF success seem limeted.

The discussion should also include a section on generalizability.

Reply: All comments raised in the discussion part have been addressed accordingly.

Minor points:

Tbl. 1: familiy income - <1000 is twice

Reply: Sorry for the incontinence, the second one has been removed. 

Gender bias, to me this does not seem to be a bias, but a gender difference in feeding practices. There is some literature on the topic and hypothesis on the reasons for this difference.

Reply: authors have reviewed other literatures and modified accordingly. 

Reviewers' comments:

Reviewer's Responses to Questions

Comments to the Author

1. Is the manuscript technically sound, and do the data support the conclusions?

Reviewer #1: Partly

2. Has the statistical analysis been performed appropriately and rigorously?

Reviewer #1: No

Reply: The procedure for selecting the candidate variables for the multivariate analysis has been stated. 

3. Have the authors made all data underlying the findings in their manuscript fully available?

Reviewer #1: No

Reply: We have included the minimal data set in the manuscript as a separate attachment called ‘supporting information.

4. Is the manuscript presented in an intelligible fashion and written in standard English?

Reviewer #1: No

Reply: The whole manuscript has been copy edited thoroughly for language usage, spelling and grammar by colleague who is expert in English.

5. Review Comments to the Author

Reviewer #1: I found that title of the manuscript was very interesting. However, the whole body of the manuscript was written in poor English. Even the language in the abstract seems a zero-draft document. The discussion is very shallow which leads to shallow conclusion. The binary and multivariate analysis should have been separately presented. Nothing was stated on how the candidate variables for the multivariate analysis were selected. What's the justification behind including all the 11 kebeles of the district? In general, this manuscript needs critical revision.

Reply: Thank you for your constructive comments. The authors have taken all the raised issues and reacted accordingly. The whole manuscript has been copy edited thoroughly for language usage, spelling and grammar by colleague who is expert in English. Literature review has been made and discussed with the finding of this study in detail. Thus, sound conclusion has been drawn. The P value ≤0.25 was taken as a cut-off point to select eligible variables for the multiple logistic regression analysis to control for potential confounders Due to the small number of kebeles in the district, all of them were considered. In general, the manuscript has been critically revised by authors 

6. PLOS authors have the option to publish the peer review history of their article (what does this mean?). If published, this will include your full peer review and any attached files.

Do you want your identity to be public for this peer review? For information about this choice, including consent withdrawal, please see our Privacy Policy.

Reviewer #1: No

---

## [Decision Letter · Decision Letter 1]

22 Jul 2021

PONE-D-20-34804R1

Infant and Young Child Feeding Practice among Mothers with Children age between 6 to 23 Months at Debrelibanos District, North Showa, Oromia Ethiopia

PLOS ONE

Dear Dr. Gemmechu,

Thank you for submitting your manuscript to PLOS ONE. After careful consideration, we feel that it has merit but does not fully meet PLOS ONE’s publication criteria as it currently stands. Therefore, we invite you to submit a revised version of the manuscript that addresses the points raised during the review process.

We look forward to receiving your revised manuscript.

Kind regards,

Julia Dratva

Academic Editor

PLOS ONE

Journal Requirements:

Additional Editor Comments (if provided):

Please address the last points raised by the reviewers.

Reviewers' comments:

Reviewer's Responses to Questions

**Comments to the Author**

1. If the authors have adequately addressed your comments raised in a previous round of review and you feel that this manuscript is now acceptable for publication, you may indicate that here to bypass the “Comments to the Author” section, enter your conflict of interest statement in the “Confidential to Editor” section, and submit your "Accept" recommendation.

Reviewer #1: (No Response)

Reviewer #2: (No Response)

2. Is the manuscript technically sound, and do the data support the conclusions?

Reviewer #1: Yes

Reviewer #2: Partly

3. Has the statistical analysis been performed appropriately and rigorously? 

Reviewer #1: Yes

Reviewer #2: I Don't Know

4. Have the authors made all data underlying the findings in their manuscript fully available?

Reviewer #1: No

Reviewer #2: Yes

5. Is the manuscript presented in an intelligible fashion and written in standard English?

Reviewer #1: Yes

Reviewer #2: No

6. Review Comments to the Author

Reviewer #1: It is appreciated that almost all the raised issues are now addressed except the following few concerns.

1. The authors have also responded that the minimal dataset is separately attached as ‘supporting information. However the dataset underlying the findings in the manuscript is still not accessible if I am not mistaken.

2. Some statements still need minor editions. For example, something is wrong in the statement written as, "In this study, mothers who made ANC follow up during pregnancy were statistically significant with appropriate IYCF practices."

I think this should be rephrased in a way that the information is clearly presented.

Reviewer #2: The manuscript is about IYCF practices and their associated factors. In general the manuscript needs extensive language and grammar editing. In the introduction section, why isn't 2016 EDHS used to give some background ? In the results section, table 4 (binary and multivariate regression analysis) is not easy to read/understand, it would be better to present results in two tables and use no more than one page per table. In the discussion section, the binary and multivariate analysis results need to be discussed with more depth.

7. PLOS authors have the option to publish the peer review history of their article (what does this mean?). If published, this will include your full peer review and any attached files.

Reviewer #1: No

Reviewer #2: No

---

## [Author Response · Author response to Decision Letter 1]

28 Jul 2021

Additional Editor Comments (if provided):

Please address the last points raised by the reviewers.

Reply: The points that had been raised by reviewers have been addressed accordingly.

4. Have the authors made all data underlying the findings in their manuscript fully available?

Reviewer #1: No

Reviewer #2: Yes

Reply for Reviewer #1: We have included the additional data set in the manuscript as a separate attachment called ‘supporting information’ as S1 and S2

5. Is the manuscript presented in an intelligible fashion and written in standard English?

Reviewer #1: Yes

Reviewer #2: No

Reply for Reviewer #2: The whole manuscript has been edited thoroughly for language usage, spelling and grammar.

6. Review Comments to the Author

Reviewer #1: It is appreciated that almost all the raised issues are now addressed except the following few concerns.

1. The authors have also responded that the minimal dataset is separately attached as ‘supporting information. However the dataset underlying the findings in the manuscript is still not accessible if I am not mistaken.

2. Some statements still need minor editions. For example, something is wrong in the statement written as, "In this study, mothers who made ANC follow up during pregnancy were statistically significant with appropriate IYCF practices."

I think this should be rephrased in a way that the information is clearly presented.

Reviewer #2: The manuscript is about IYCF practices and their associated factors. 

In general the manuscript needs extensive language and grammar editing. In the introduction section, why isn't 2016 EDHS used to give some background ? In the results section, table 4 (binary and multivariate regression analysis) is not easy to read/understand, it would be better to present results in two tables and use no more than one page per table. In the discussion section, the binary and multivariate analysis results need to be discussed with more depth.

Reply for Reviewer #1: Thank you for your constructive comments. We have included the additional data set in the manuscript as a separate attachment called ‘supporting information’ i.e. S2. Authors have also edited statements that need editions so that it can be clearly understood by readers clearly. 

Replay for Reviewer #2: Thank you for taking the time to give your valuable feedback. To address the concerns identified, the entire manuscript has been thoroughly edited for language usage, spelling, and grammar. Secondly, one paragraph in the introductory section has been dedicated to providing some background information utilizing the 2016 EDHS. Thirdly, the authors improved the design and format of Table 4 in the results section to make it more understandable.Finally, the authors have added depth the discussion of multivariate analysis in the discussion section. However, related to binary analysis, it was already adjusted after considering all variables with P-value ≤0.25 in binary analysis for the multivariate logistic regression analysis to control for potential confounders.

Thank you once again for your constructive comments!

---

## [Editor Report · Decision Letter 2]

31 Aug 2021

PONE-D-20-34804R2

Infant and young child feeding practice among mothers of children age 6 to 23 months in Debrelibanos district, North Showa zone, Oromia region, Ethiopia

PLOS ONE

Dear Dr. Mekonnen,

Thank you for submitting your manuscript to PLOS ONE. After careful consideration, we feel that it has merit but does not fully meet PLOS ONE’s publication criteria as it currently stands. Therefore, we invite you to submit a revised version of the manuscript that addresses the points raised during the review process.

We look forward to receiving your revised manuscript.

Kind regards,

Julia Dratva

Academic Editor

PLOS ONE

Journal Requirements:

Additional Editor Comments (if provided):

The paper has largely improved. Thank you for providing another revision.

Please address the following points:

1. There seems to be a misunderstanding with regard to "data availability" - the authors added supplemental material S1 and S2 which are actually questionnaires/ questionnaire items used in the study. Data availability refers to the data being available on a data server or such. Please address, if it is possible or foreseen to publish the data once the paper is published. If not, please explain. Please see: https://journals.plos.org/plosone/s/data-availability

2. The first paragraph of the discussion is mainly a repetition of results already found in the result section even with percentages. Please rewrite the first paragraph summing up the main results respectively the main insights, without this detail. You can of course repeat results in the discussion, when you put them into comparison with literature or when interpreting them.

3. Table 4: please separate COR/AOR values from the CI values: 12.859(6.135,26.955) => 12.859 (6.135,26.955) and provide the same number of digtis after the comma for all values. Add defintion of COR and AOR to the legend.
---

## [Author Response · Author response to Decision Letter 2]

6 Sep 2021

1. There seems to be a misunderstanding with regard to "data availability" - the authors added supplemental material S1 and S2 which are actually questionnaires/ questionnaire items used in the study. Data availability refers to the data being available on a data server or such. Please address, if it is possible or foreseen to publish the data once the paper is published. If not, please explain. Please see: https://journals.plos.org/plosone/s/data-availability

Reply: Thank you for your constructive comment. Authors have now added Data Set used in the study as “S1 Data (SAV)” in addition to questionaries used in the study which has been submitted as “S 1 File (DOCX)” under title “Supporting information” right after Conclusion in manuscript. All relevant data are declared to be within the manuscript and its Supporting Information files during submission process.

2. The first paragraph of the discussion is mainly a repetition of results already found in the result section even with percentages. Please rewrite the first paragraph summing up the main results respectively the main insights, without this detail. You can of course repeat results in the discussion, when you put them into comparison with literature or when interpreting them.

Reply: Modification has been made accordingly. 

3. Table 4: please separate COR/AOR values from the CI values: 12.859(6.135,26.955) => 12.859 (6.135,26.955) and provide the same number of digits after the comma for all values. Add definition of COR and AOR to the legend.

Reply: Modification has been made accordingly.

---

## [Editor Report · Decision Letter 3]

10 Sep 2021

Infant and young child feeding practice among mothers of children age 6 to 23 months in Debrelibanos district, North Showa zone, Oromia region, Ethiopia

PONE-D-20-34804R3

Dear Dr. Mekonnen,

We’re pleased to inform you that your manuscript has been judged scientifically suitable for publication and will be formally accepted for publication once it meets all outstanding technical requirements.

Kind regards,

Julia Dratva

Academic Editor

PLOS ONE

Additional Editor Comments (optional):

Thank you for the revision of your paper and provision of the data used in this mansucript.

The journal will need to assess if this format is correct. I cannot.
---

## [Editor Report · Acceptance letter]

16 Sep 2021

PONE-D-20-34804R3 

Infant and young child feeding practice among mothers of children age 6 to 23 months in Debrelibanos district, North Showa zone, Oromia region, Ethiopia 

Dear Dr. Mekonnen:

I'm pleased to inform you that your manuscript has been deemed suitable for publication in PLOS ONE. Congratulations! Your manuscript is now with our production department. 

Kind regards, 

on behalf of

Dr. Julia Dratva 

Academic Editor

PLOS ONE